

# Identification and validation of HELLS (Helicase, Lymphoid-Specific) and ICAM1 (Intercellular adhesion molecule 1) as potential diagnostic biomarkers of lung cancer

Wei Zhu[1],[*], Lin Lin Li[1],[*], Yiyan Songyang[1], Zhan Shi[2] and Dejia Li[1]

[1] Department of Occupational and Environmental Health, Wuhan University, School of Health Science, Wuhan, Hubei, China
[2] Human Biology Program, University of Toronto, Toronto, ON, Canada
[*] These authors contributed equally to this work.

Corresponding author
Dejia Li, djli@whu.edu.cn

## ABSTRACT

Although lung cancer is one of the greatest threats to human health, its signaling pathway and related genes are still unknown. This study integrates data from three groups of people to study potential key candidate genes and pathways related to lung cancer. Expression profiles (GSE18842, GSE19188 and GSE27262), including 162 tumor tissue and 135 adjacent normal lung tissue samples, were integrated and analyzed. Differentially expressed genes (DEGs) and candidate genes were identified, their expression pathways were analyzed, and the diethylene glycol-related protein–protein interaction (PPI) network was analyzed. We identified 232 shared DEGs (40 upregulated and 192 down-regulated) from the three GSE datasets. The DEGs were clustered according to function and signaling pathway for significant enrichment analysis. In total, 129 nodes/DEGs were identified from the DEG PPI network complex. An improved prognosis was associated with increased Helicase, Lymphoid-Specific (HELLS) and decreased Intercellular adhesion molecule 1 (ICAM1) mRNA expression in lung cancer patients. In conclusion, we used integrated bioinformatics analysis to identify candidate genes and pathways in lung cancer to show that HELLS and ICAM1 might be the key genes related to tumorigenesis or tumor progression in lung cancer. Additional studies are needed to further explore the involved functional mechanisms.

Subjects Bioinformatics, Molecular Biology, Oncology, Respiratory Medicine
Keywords Lung cancer, Bioinformatics analysis, Prognosis, HELLS, ICAM1

## INTRODUCTION

Lung cancer is one of the most leading causes of cancer-related deaths worldwide. In 2016, there were more than 220,000 diagnoses and nearly 158,000 deaths from lung cancer in the United States alone (Siegel, Miller & Jemal, 2016). There are two main histological types of lung cancer: non-small cell lung cancer (NSCLC) and small cell lung cancer. The former accounts for about 85% of all lung cancer, including squamous cell carcinoma, adenocarcinoma and large cell carcinoma (Siegel, Miller & Jemal, 2018). Although
significant progress has been made in diagnosis and treatment methods in the last 5 years, the overall survival (OS) rate of lung cancer is still less than 15% (*Bironzo, Passiglia & Novello, 2019*). Therefore, the molecular mechanisms involved in the development of lung cancer should be studied and clarified in order to improve survival rates.

Gene chip, or gene expression profile, is a genetic detection technology that is particularly useful for screening differential gene expression since it has the ability to rapidly detect all genes expressed in the same sample after the time of sampling (*Vogelstein et al., 2013*). The widespread use of gene chips has generated a large amount of core slice data stored in public databases. Useful information has been provided from integrating and analyzing this data. In recent years, a large number of microarray data analyses on lung cancer have been conducted, and hundreds of differentially expressed genes (DEGs) have been identified. However, due to the heterogeneity of the tissue or samples in the existing studies, the results were limited or inconsistent. In cancer tissues, heterogeneity means that cells with different gene mutations may have different biological characteristics. The clinical diagnosis of cancer by pathologists usually relies on limited samples of cancer tissue that do not represent heterogeneity, either between or within patients (*Bedard et al., 2013*; *De Sousa & Carvalho, 2018*).

As a result, reliable biomarkers have not been found in lung cancer. A novel approach to address these shortcomings is to incorporate a comprehensive bioinformatics approach to expression profiling techniques, which is the approach we adopted in this study.

We first downloaded three original microarray datasets, GSE18842 (*Sanchez-Palencia et al., 2011*), GSE19188 (*Hou et al., 2010*) and GSE27262 (*Wei et al., 2014*, *2012*), from the NCBI Gene Expression Synthesis Database (NCBI–GEO, https://www.ncbi.nlm.nih.gov/geo) (*Barrett et al., 2005*). Data were obtained from 162 lung cancer cases and 135 adjacent normal tissues. The principles of our dataset selection were as following: (1) the sample size was greater than 50; (2) the samples were all from lung cancer patients and paracancer tissues; (3) these patients had not undergone any other drug intervention; and (4) the purposes of carrying out the gene chip or RNA-seq were to compare and analyze the RNA expression differences between lung cancer patients and paracancer tissues. We screened the corresponding DEGs according to the data processing standards of the Morpheus website and used DAVID, Cytoscape, Metascape (http://metascape.org/) (*Zhou et al., 2019*), UCSC (https://genome.ucsc.edu/) (*Haeussler et al., 2019*), cBioportal (*Gao et al., 2013*), BioCyc (http://biocyc.org) (*Latendresse, Paley & Karp, 2012*) and Panther (http://www.pantherdb.org) to perform gene ontology and pathway enrichment analysis. We also developed a comprehensive DEG protein–protein interaction (PPI) network and module analysis to identify the central gene of lung cancer using the Search Tool for the Retrieval of Interacting Genes/Proteins database (STRING, http://string-db.org). To identify the central lung cancer genes using string (http://string-db.org), we also developed a comprehensive DEG PPI network and module analysis. Helicase, Lymphoid-Specific (HELLS) and Intercellular adhesion molecule 1 (ICAM1) were identified, and their biological functions and key pathways were enriched to ascertain more accurate and practical biomarkers for the early diagnosis, individualized prevention, and treatment of lung cancer. Finally, we analyzed the expression of HELLS and ICAM1 in lung cancer

patients to determine their expression patterns, potential functions, and different prognostic values.

## MATERIALS AND METHODS

### Microarray data analysis and identification of DEGs

We obtained lung cancer and adjacent tissue gene expression profiles for GSE18842, GSE19188 and GSE27262 from NCBI to GEO, which is a free microarray/gene database repository of high throughput gene expression data. Microarray data for GSE18842 were based on the GPL570 platform ((HG-U133_Plus_2) Affymetrix Human Genome U133 Plus 2.0 Array), and included 46 tumors and 45 controls (submission date: 2 November 2009) (*Latendresse, Paley & Karp, 2012*). GSE19188 data were based on the GPL570 platform ((HG-U133_Plus_2) Affymetrix Human Genome U133 Plus 2.0 Array), and included 91 tumor and 65 adjacent normal lung tissue samples (submission date: 25 November 2009) (*Hou et al., 2010*). GSE27262 data were based on the GPL570 platform ((HG-U133_Plus_2) Affymetrix Human Genome U133 Plus 2.0 Array), and included 25 pairs of tumor and adjacent normal tissues from LUAD patients (submission date: 11 February 2011) (*Wei et al., 2014*, *2012*). We selected these three datasets for integrated analysis and identified DEGs using a classical $t$ test. The adjusted $p$ values (adj. $p$) were utilized to correct the occurrence of false-positive results using the Benjamini and Hochberg false discovery rate method by default. In the present study, statistically significant DEGs were defined using values of adj. $p < 0.05$ and [logFC] > 1 as cutoff criteria (*The Gene Ontology Consortium, 2015*; *Ashburner et al., 2000*; *Lebrec et al., 2009*).

### Gene ontology and pathway enrichment analysis

Metascape (http://metascape.org/) is an online analysis tool for extracting comprehensive biometric information from huge lists of candidate genes. It not only performs typical genetic terminology enrichment analysis, but also visualizes the relationship between genomic terms, searches for interesting and related genes or terms, and dynamically views genes from their biological functions and pathways. GO analysis and the Kyoto Genomics and Genomics Encyclopedia (KEGG) path analysis were conducted on the selected DEG using the Metascape tool. The enrichment score [−log 10 ($p$-value)] was significantly ranked with a $p$-value <0.01 as the cutoff criterion.

### Integration of PPI network complex identification

We developed a DEG-encoded protein and PPI network using STRING (http://string-db.org) (*Szklarczyk et al., 2019*). The PPI network was constructed using Cytoscape software (version 3.7.1) to analyze the interactions between candidate DEG-encoded proteins in lung cancer (*Kohl, Wiese & Warscheid, 2011*). The Node Analyzer was calculated using the Network Analyzer plug-in, which reveals the number of connections used to filter the PPI hub genes. The corresponding protein identified at the central node may be a core protein and key candidate gene with important physiological regulatory functions.

### Identification and clinical significance of central genes

Hub genes were identified using CythopCAE's CyoHubBA toolkit. The top 10 central genes with less than 10 degrees were selected. Hierarchical clustering of hub genes in the Cancer Genome Atlas (TCGA) database was constructed using the UCSC Cancer Genome Browser (https://genome.ucsc.edu/). Biological process analysis of the hub gene was performed utilizing the Cytoscape Bionetwork Gene Oncology Tool (BiNGO) plug-in. The frequency of gene changes was assessed using the cBioportal online database (http://www.cbioportal.org/). PPI networks were built using STRING.

### Prognosis analysis using Kaplan–Meier plots

The TCGA online database, which contains gene expression data and survival information for lung cancer patients and sequencing and pathological data for 30 different cancers, was utilized to evaluate the prognostic value of DEG expression (*The Cancer Genome Atlas Network, 2012*). To analyze the OS of patients with lung cancer, 545 patients were assigned to either two groups (high and low expression) or three groups (high, medium and low expression) by median expression and were assessed using Kaplan–Meier survival analysis with hazard ratio, 95% confidence intervals, and log-rank $p$-value. Only the JetSet best probe set of DEGs was selected to obtain a Kaplan–Meier plot, with the risk number shown below the main plot.

### Lung cancer samples

Lung cancer and adjacent normal tissues were obtained after surgical resection of patients with NSCLC being treated at Wuhan University Affiliated Hospital, Hubei Provincial People's Hospital, and Zhongnan Hospital of Wuhan University, China. Informed consent was received by each patient. The study was approved by Wuhan University's Institute of Ethics with certificate number 2018001. It is presumed that informed consent had been obtained for all datasets used from the published literature.

### RT-qPCR

Total RNA was extracted from lung cancer cells and tissues using Trizol reagent (Invitrogen; Thermo Fisher Scientific, Inc., Waltham, MA, USA). A reverse transcription kit (Vazyme Biotech Co., China) was used to reverse transcribe RNA and mRNA expression was evaluated using the $2^{(\Delta\Delta Ct)}$ method. Relative gene expression was then calculated and normalized to endogenous glyceraldehyde 3-phosphate dehydrogenase (GAPDH). The primers of GAPDH, HELLS, ICAM1 were purchased from Qingke Biotechnology Co., Ltd., China. The forward and reverse primer sequences are as follows: GAPDH F: 5′-CCTTCCGTGTCCCCACT-3′ and GAPDH R: 5′-GCCTGCTTCACCACC TTC-3′, HELLS F: 5′-CCCTCCTTTCTTCTAGTAATGCAGTT-3′ and HELLS R: 5′-CCCAATCTCTCCCCATGAAAA-3′; ICAM1 F: 5′-GAACCCATTGCCCGAGCTC A-3′ and ICAM1 R: 5′-TGACAGTCACTGATTCCCCGAT-3′.

### Cell culture

Human NSCLC cell lines A549, H1299, PC9 and HCC827 and the normal lung epithelial cell line BEAS2B were all bought from the Shanghai Cell Bank of the Chinese Academy of

Sciences and were cryopreserved in liquid nitrogen tanks. The cells were cultured in DubCo's MudieEdEdE medium (Hyo Corporation, Salt Lake City, UT, USA) with 10% fetal bovine serum (GiBCO Co., Grand Island, NY, USA) and incubated at 37 °C in a 5% $CO_2$ atmosphere.

## Statistical analysis

Experimental data were recorded in Excel and analyzed using GraphPad Prism 7. The results were analyzed by a two-tailed $t$ test. Values of $p < 0.05$ were considered significant differences. Data were expressed as mean ± standard error.

## RESULTS

### Identification of DEGs in lung cancer

Lung cancer and adjacent normal tissue gene expression profiles for GSE18842, GSE19188 and GSE27262 were obtained from NCBI–GEO. Microarray data for GSE18842 comprised 46 tumors and 45 controls (*Sanchez-Palencia et al., 2011*). GSE21815 data comprised 91 tumor and 65 adjacent normal lung tissue samples. GSE27262 data included 25 paired tumor and adjacent normal tissue samples. In total, 2,042, 1,424 and 1,142 DEGs were extracted from the GSE18842, GSE19188 and GSE27262 expression profile datasets, respectively, using adj. $p < 0.05$ and [logFC] > 1 as cutoff criteria. A total of 232 uniformly expressed genes were identified from the three profile datasets using integrated analysis (Figs. 1 and 2). When compared to normal lung tissue, lung cancer tissues included 40 up-regulated genes and 192 down-regulated genes (Table 1).

### Functional and pathway enrichment analysis of DEGs

The functions and pathways of the candidate DEGs were predicted using the Metascape Database (Table 2) (*Geiman, Durum & Muegge, 1998*). There were 18 terms and two pathways involved in the DEGs enrichment analysis (Fig. 1A), and the DEGs were mainly enriched during developmental growth, embryonic morphogenesis, cell-substrate junction assembly, renin secretion, regulation of cell adhesion, myeloid leukocyte activation, mesenchyme development, assembly of cellular components involved in morphogenesis and muscle system processes, in the cell surface receptor signaling pathway involved in cell–cell signaling, positive regulation of protein transport, endodermal cell differentiation, negative regulation of cell proliferation, hemostasis and ameboidal-type cell migration. The GO function and KEGG pathway enrichment analysis of candidate DEGs are shown in Fig. 1. The enriched terms were closely connected with each other and clustered into intact networks (Fig. 1B). These results indicate that most DEGs are significantly enriched during cardiomyocyte proliferation, protein binding, and the positive regulation of cell membranes.

### DEGs modular analysis with PPI network and hub gene identification

Protein interaction networks have proven to be powerful tools for predicting new essential genes in specific signal transduction pathways. Using the STRING online database and Cytoscape software 13, a total of 232 DEGs were filtered into the PPI network complex.

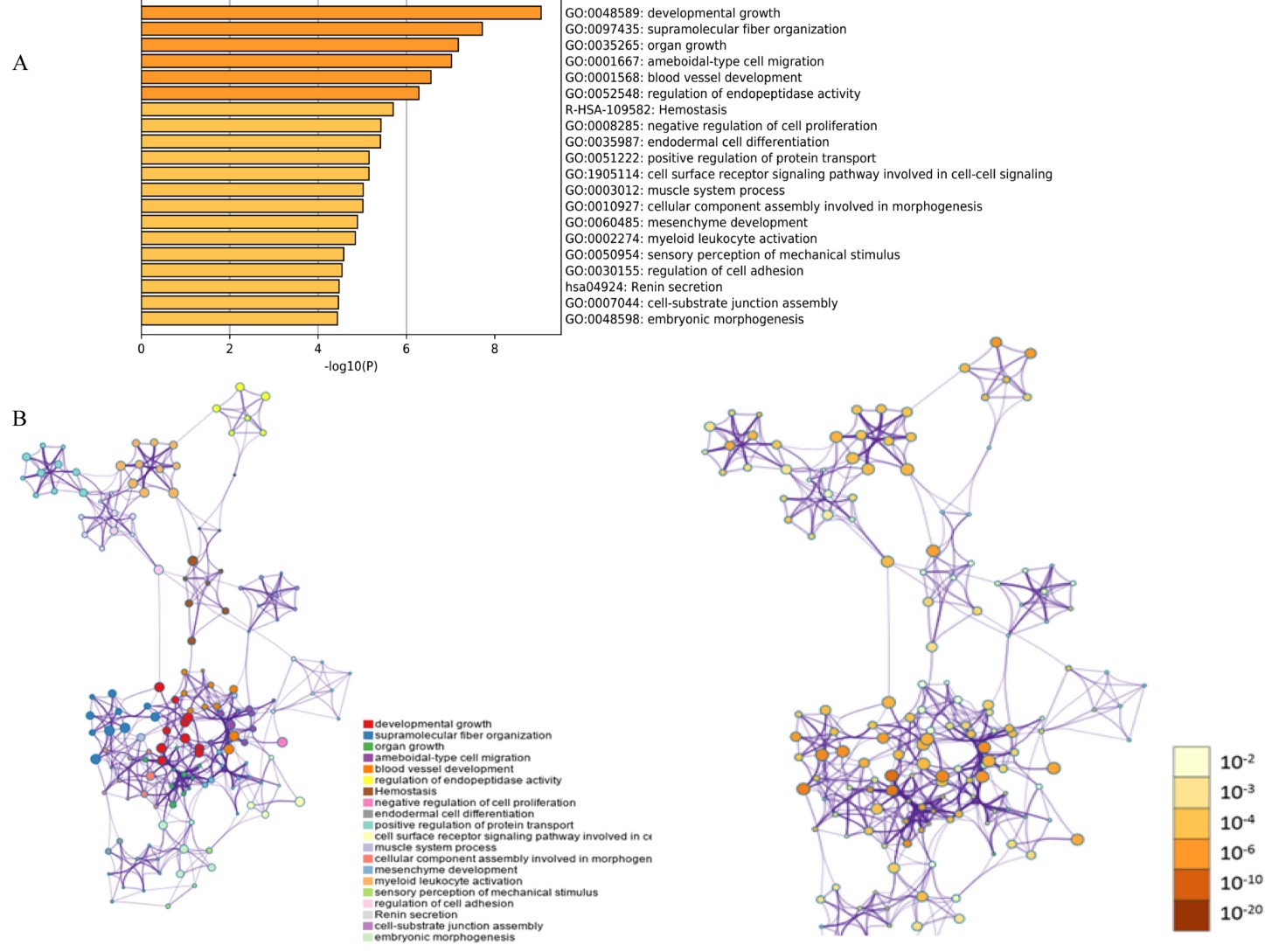

**Figure 1 Functional and pathway enrichment analysis of DEGs.** (A) GO terms and KEGG pathway were presented, and each band represents one enriched term or pathway colored according to the −log 10 *p* value. (B) Network of the enriched terms and pathways. Nodes represent enriched terms or pathways with node size indicating the number of DEGs involved in. Nodes sharing the same cluster are typically close to each other, and the thicker the edge displayed, the higher the similarity is represented.

The PPI network of DEGs was constructed with the most important module obtained using the Cytoscape (Figs. 2B and 2C). Using the Metascope to analyze the functions of the genes involved in this module, we found that the functions mainly focused on cell division and pre-mitotic stage and mitosis cell cycle transition (Fig. 3).

## Hub gene selection and analysis

The top 10 node degree genes were MAD2L1, POLQ, HELLS, ANLN, BIRC5, ATAD2, CCNB2, PTK2, ICAM1 and ITGAX (Table 3). A network of hub genes and their co-expressed genes were analyzed using the cBioPortal online platform (Fig. 4A). The analysis of the biological processes of the hub genes, which was constructed using

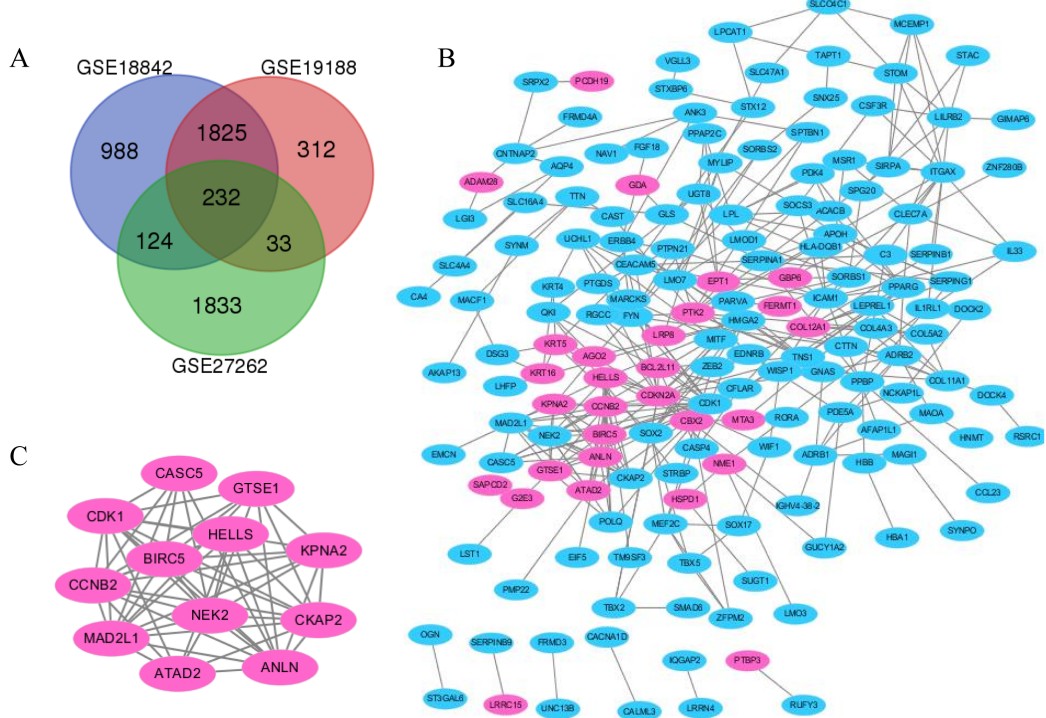

**Figure 2 The distribution of differential genes between GSE18842, GSE19188 and GSE27262.**
(A) DEGs were selected with $p < 0.05$ and [log FC] > 1 among the mRNA expression profiling sets.
(B and C) The PPI network of DEGs was constructed using Cytoscape (upregulated genes are marked in
light red; downregulated genes are marked in light blue).

**Table 1 Two hundred thirty-two differentially expressed genes (DEGs) were screened from three profile datasets.**

| DEGs | Genes symbol |
| --- | --- |
| Upregulated (40) | G2E3, GDA, EPT1, KPNA2, IGH, ANLN, BIRC5, HSPD1, PTK2, BC017398, MIR3934, SAPCD2, IGHD, AGO2, GTSE1, CBX2, PTBP3, ADAM28, CCNB2, LRP8, NFE2L3, KRT16, IGHG1, NME1, COL12A1, EYA2, LEPREL4, LRRC15, BCL2L11, ATAD2, MTA3, HIST1H2BG, PCDH19, SLC1A4, HELLS, GBP6, FERMT1, KRT5, HOXC6, CDKN2A |
| Downregulated (192) | PFKP, ERG, HIGD1B, VGLL3, IPW, MAOA, CDK1, RSRC1, SPTBN1, SNX25, UNC13B, PPBP, QKI, SPG20, MAD2L1, SORBS2, CCM2L, WIF1, GIMAP6, SOCS3, MAGI1, LMO7, CCL23, PCDP1, OGN, KRT4, SFTA3, CEACAM5, PDE5A, SLC16A4, PDZD2, WISP1, HBB, ITGAX, TM9SF3, LPL, COL11A1, ODF3B, CASP4, ROR1, SYNM, UGT8, FKBP11, SESTD1, SLC4A4, RP699M1.2, CTTN, NEK2, SMAD6, MEF2C, ERBB4, RP115C23.1, MACF1, MAGEA10, MAGEA5, ITIH5, SIGLEC17P, TBX5, PARVA, PPAP2C, AQP4, SLC47A1, SERPINA1, COL4A3, IL1RL1, MCEMP1, CYP4V2, TRPV2, STOM, KIAA1244, EDNRB, ST3GAL6, SOX17, TNS1, CAMK2N1, POLQ, CACNA1D, RGS5, PTGDS, GAGE12B, EIF5, SERPINB1, PTPN21, CST6, STRBP, NAV1, SHROOM3, CNTNAP2, ZNF280B, LMO3, MBIP, IL33, ARHGAP6, RNF125, CYP2B6, ANK3, DAPL1, KCTD1, TACC2, MITF, LILRB2, HOXA1, CSF3R, LOC643733, HNMT, GNAS, SLC27A3, SERPINB9, C3, SERPING1, AFAP1L1, SULT1A2, ZFPM2, SEC63, ADRB1, SVEP1, FYN, COL5A2, LOC101928198, PLAC9, MSR1, LST1, DOCK4, FRMD4A, KLF9, PDK4, EMCN, TSPAN12, CA4, SRPX2, SIRPA, APOH, CLEC7A, NCKAP1L, LHFP, GLS, CFLAR, ACACB, RUFY3, SOBP, PMP22, P2RX7, LEPREL1, LPCAT1, SOX2, IQGAP2, OTUD1, FRMD3, DOCK2, BTNL9, UCHL1, CLEC2B, TBX2, TMEM237, PPARG, HLADQB1, LMOD1, SUGT1, LRRN4, RGCC, ADRB2, CMAHP, SEMA6A, HMGA2, CCDC68, SREK1IP1, MYLIP, DOCK9, MARCKS, RORA, SORBS1, GUCY1A2, STXBP6, STX12, CASC5, CALML3, CKAP2, ICAM1, FGF18, ZEB2, DSG3, LGI3, TTN, AKAP13, SLC34A2, STAC, TAPT1, SEMA5A, SYNPO, CAST, SLCO4C1, HBA1 |

**Table 2 Pathway and process enrichment analysis.**

| GO | Category | Description | Count | % | log 10 (p) |
|---|---|---|---|---|---|
| Upregulated | | | | | |
| M236 | Canonical Pathways | PID DELTA NP63 PATHWAY | 4 | 10.53 | −6.05 |
| M176 | Canonical Pathways | PID FOXM1 PATHWAY | 3 | 7.89 | −4.47 |
| R-HSA-5693606 | Reactome Gene sets | DNA Double Strand Break Response | 3 | 7.89 | −3.62 |
| M66 | Canonical Pathways | PID MYC ACTIV PATHWAY | 4 | 7.89 | −3.58 |
| GO:0007160 | GO Biological Processes | Ell-matrix adhesion | 5 | 10.53 | −3.37 |
| GO:1903828 | GO Biological Processes | Negative regulation of cellular protein localization | 3 | 7.89 | −3.16 |
| GO:0008637 | GO Biological Processes | Apoptotic mitochondrial changes | 3 | 7.89 | −3.01 |
| R-HSA-9006925 | Reactome Gene sets | Intracellular signaling by second messenger | 3 | 10.53 | −2.89 |
| GO:0042742 | GO Biological Processes | Defense response to bacterium | 4 | 10.53 | −2.78 |
| GO:0051301 | GO Biological Processes | Cell division | 5 | 13.16 | −2.65 |
| GO:0017038 | GO Biological Processes | Protein import | 3 | 7.89 | −2.48 |
| GO:0048872 | GO Biological Processes | Homeostasis of number of cells | 3 | 7.89 | −2.15 |
| Downregulated | | | | | |
| GO:0048589 | GO Biological Processes | Developmental growth | 23 | 12.17 | −8.59 |
| GO:0030036 | GO Biological Processes | Actin cytoskeleton organization | 22 | 11.64 | −8.04 |
| GO:0001568 | GO Biological Processes | Blood vessel development | 22 | 11.64 | −6.79 |
| GO:0002274 | GO Biological Processes | Myeloid leukocyte activation | 19 | 10.05 | −6.01 |
| GO:0052548 | GO Biological Processes | Regulation of endopeptidase activity | 15 | 7.94 | −5.93 |
| GO:0060485 | GO Biological Processes | Mesenchyme development | 12 | 6.35 | −5.70 |
| R-HSA-109582 | Reactome Gene sets | Hemostasis | 18 | 9.52 | −5.67 |
| GO:0010927 | GO Biological Processes | Cellular component assembly involved in morphogenesis | 8 | 4.23 | −5.60 |
| GO:0003012 | GO Biological Processes | Muscle system process | 15 | 7.94 | −5.36 |
| GO:1905114 | GO Biological Processes | Cell surface receptor signaling pathway involved in cell–cell signaling | 17 | 8.99 | −5.08 |
| GO:0008285 | GO Biological Processes | Negative regulation of cell proliferation | 19 | 10.05 | −5.00 |
| hsa04924 | KEGG Pathway | Renin secretion | 6 | 3.17 | −4.92 |
| GO:0048588 | GO Biological Processes | Developmental cell growth | 10 | 5.29 | −4.86 |
| GO:0010884 | GO Biological Processes | Positive regulation of lipid storage | 4 | 2.12 | −4.62 |
| GO:0003013 | GO Biological Processes | Circulatory system process | 15 | 7.94 | −4.55 |
| GO:0051345 | Reactome Gene Sets | Posithive regulation of hydrolase activity | 18 | 9.52 | −4.49 |
| GO:0043408 | GO Biological Processes | Regulation of MAPK cascade | 18 | 9.52 | −4.44 |
| R-HSA-1247673 | GO Biological Processes | Erythrocytes take up oxygen release | 3 | 1.59 | −4.42 |
| GO:0099612 | GO Biological Processes | Protein localization to axon | 3 | 1.59 | −4.42 |
| GO:0045444 | GO Biological Processes | Fat cell differentiation | 9 | 4.76 | −4.21 |

plug-in BiNGO, is shown in Fig. 4C. Hierarchical clustering indicated that the hub genes could differentiate cancer samples from noncancerous samples (Fig. 4B). These genes may play a significant role in lung cancer development.

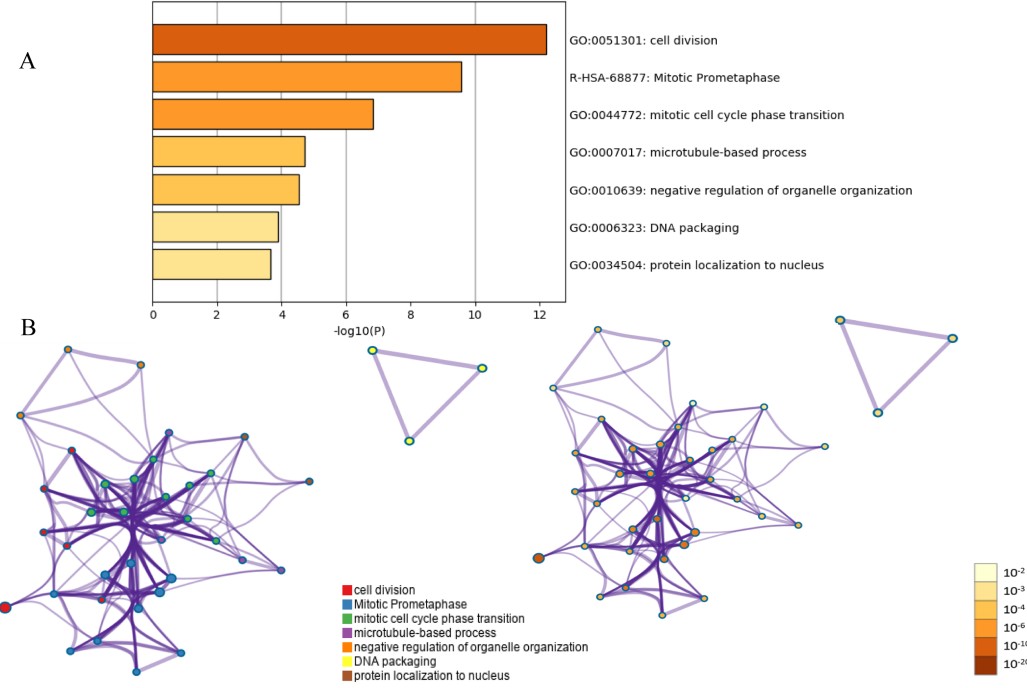

**Figure 3 Functional and pathway enrichment analysis of the PPI module.** (A) GO terms and KEGG pathway were presented, and each band represents one enriched term or pathway colored according to the −log 10 $p$ value. (B) Network of the enriched terms and pathways. Nodes represent enriched terms or pathways with node size indicating the number of DEGs involved in.

**Table 3 Functional roles of 10 hub genes.**

| Genes symbol | Full name | Function |
| --- | --- | --- |
| MAD2L1 | Mitotic arrest deficient 2 like 1 | Preventing the onset of anaphase |
| POLQ | DNA polymerase θ | Alternative nonhomologous end joining |
| HELLS | Helicase, lymphoid specific | DNA strand separation |
| ANLN | Aniline actin binding protein | Cell growth and migration |
| BIRC5 | Bucovina IAP repeat containing 5 | Preventing apoptotic cell death |
| ATAD2 | ATPase family AAA domain containing 2 | Chaperone-like functions |
| CCNB2 | Cyclin B2 | The cell cycle regulatory machinery |
| PTK2 | Protein tyrosine kinase 2 | Cytoplasmic protein tyrosine kinase |
| ICAM1 | Intercellular adhesion molecule 1 | Endothelial cells and cells of the immune system |
| ITGAX | Integrin subunit alpha X | Encoding the integrin alpha X chain protein |

## Association between HELLS and ICAM1 expression and prognoses in lung cancer patients

We used the TCGA website to further explore the 10 central genes related to survival of lung cancer patients. According to curve and logarithmic rank test analysis, the elevated

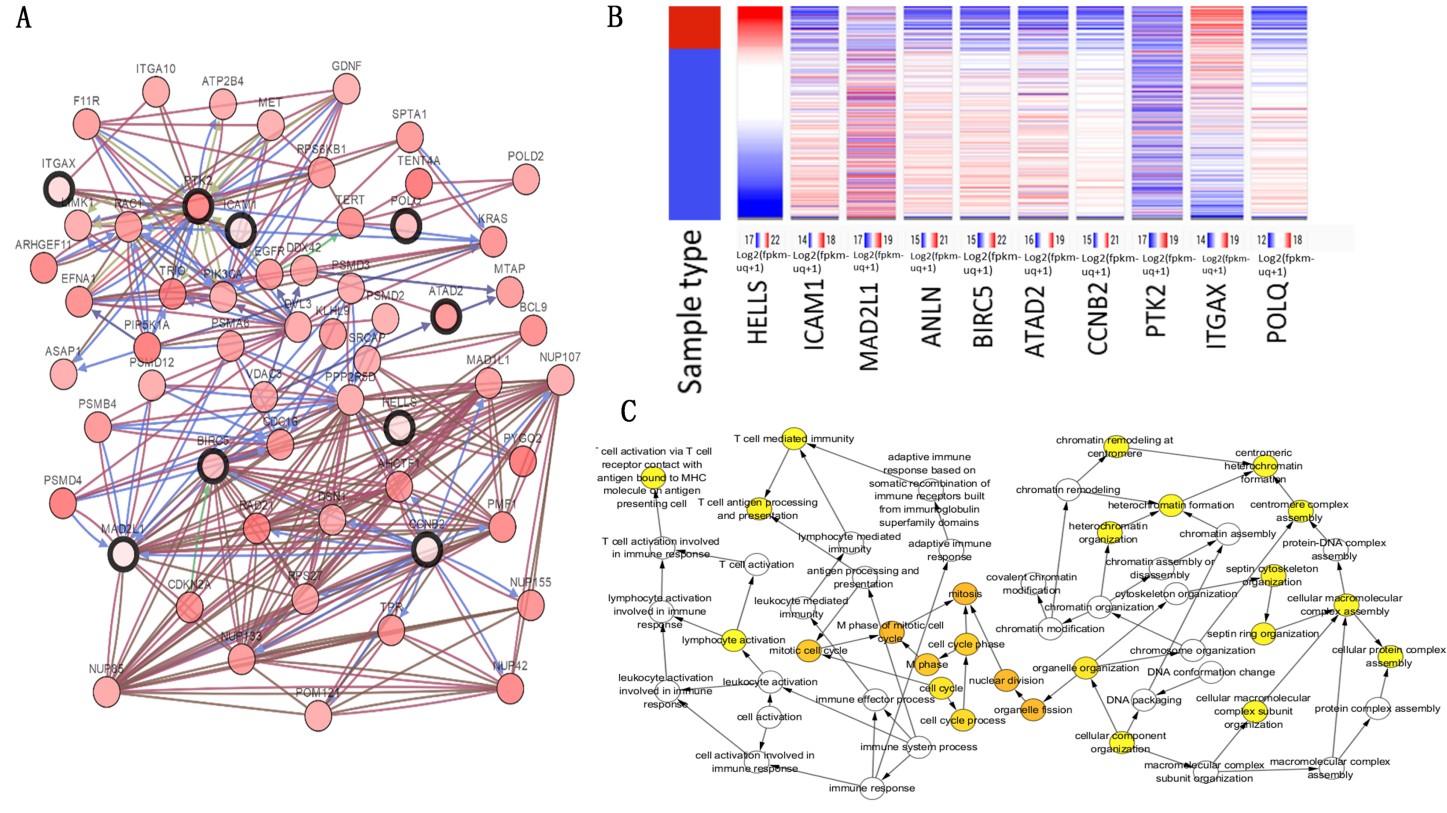

**Figure 4 Hub gene selection and analysis.** (A) Hub genes and their co-expression genes were analyzed using cBioPortal. Nodes with bold black outline represent hub genes. Nodes with thin black outline represent the co-expression genes. (B) Hierarchical clustering of hub genes was constructed using UCSC. (C) The biological process analysis of hub genes was constructed using BiNGO.

level of HELLS mRNA correlated significantly with OS difference in lung cancer patients (Fig. 5). Interestingly, lower ICAM1 levels also indicated poor prognoses in lung cancer patients (Fig. 6). After assessing the mRNA levels of ICAM1 and HELLS using the Oncomine online database (https://www.oncomine.org/resource/login.html) (*Rhodes et al., 2004*) (Figs. 7A and 7B), it was indicated that ICAM1 was down-regulated in lung cancer across five differet studies. Furthermore, HELLS expression was upregulated in lung cancer tumors. After HELLS and ICAM1 were identified from these 10 central genes, Gene Expression Profiling interactive analysis (GEPIA; http://gepia.cancer-pku.cn/) was used to validate the selected upregulated and downregulated genes (*Tang et al., 2017*). The GEPIA analysis includes data from TCGA and the Genotype Tissue Expression, and provides online gene expression level analysis, survival analysis, and tumor staging analysis for 33 types of cancers, including lung adenocarcinoma (LUAD) and lung squamous cell carcinoma (LUSC). The mRNA level of HELLS was evaluated in lung cancer using GEPIA analysis, and the expression of HELLS in lung cancer tissues was significantly higher than in adjacent tissues ($p < 0.05$) (Fig. 5G). The expression of ICAM-1 in LUSC was significantly lower than in paracancer tissues ($p < 0.05$) (Fig. 6G).

A PPI network of ICAM1 and HELLS was constructed using the STRING database. The results indicated that HELLS was associated with other genes in the minichromosome

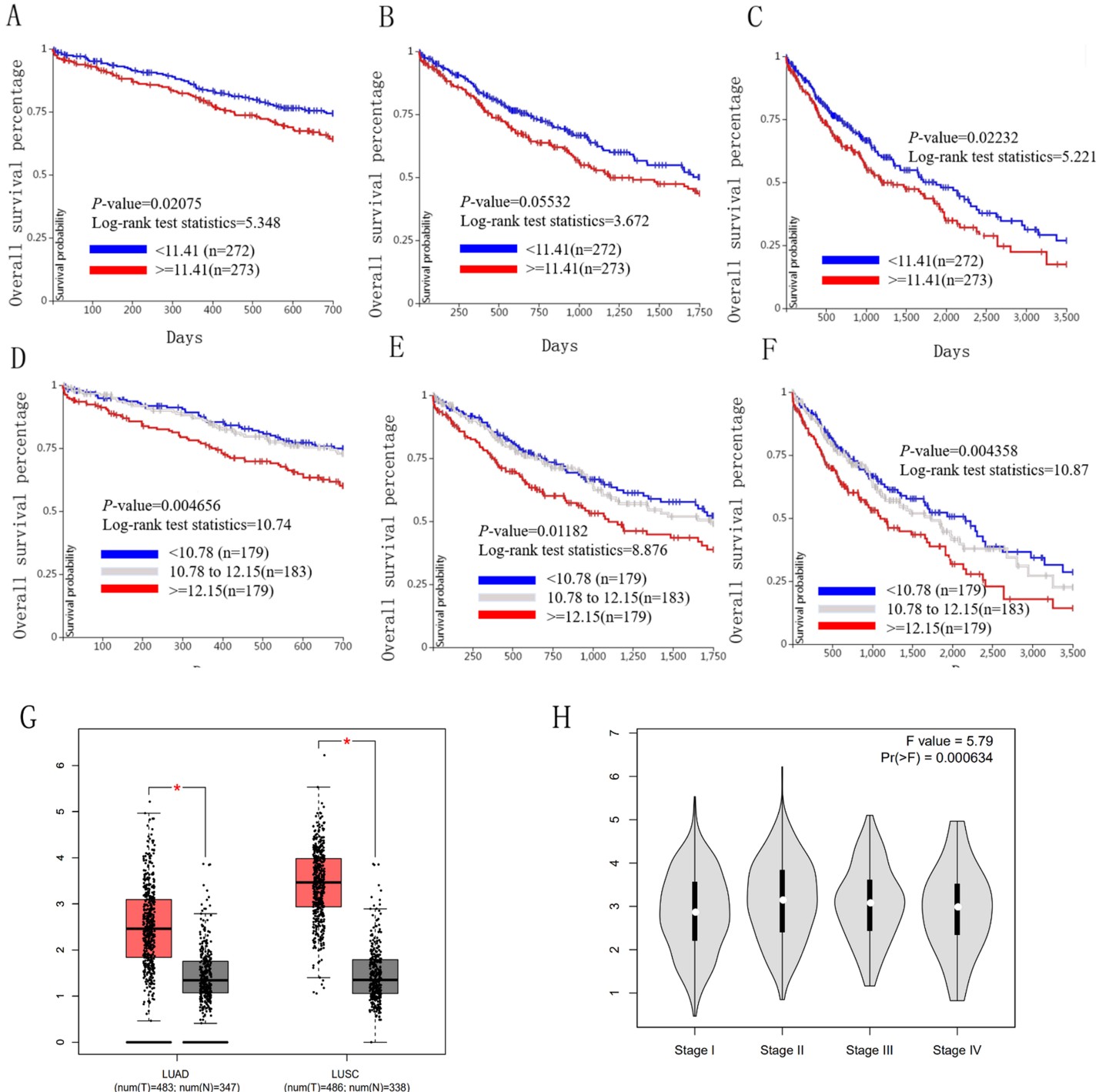

**Figure 5  Overall survival and disease-free survival analyses of different HELLS expression lung cancer patients.** (A–F) Overall survival and disease-free survival analyses of the HELLS were performed in TCGA online website. (G and H) The mRNA level of HELLS was evaluated in lung cancer using GEPIA analysis, $p < 0.05$ was considered to indicate a statistically significant difference.

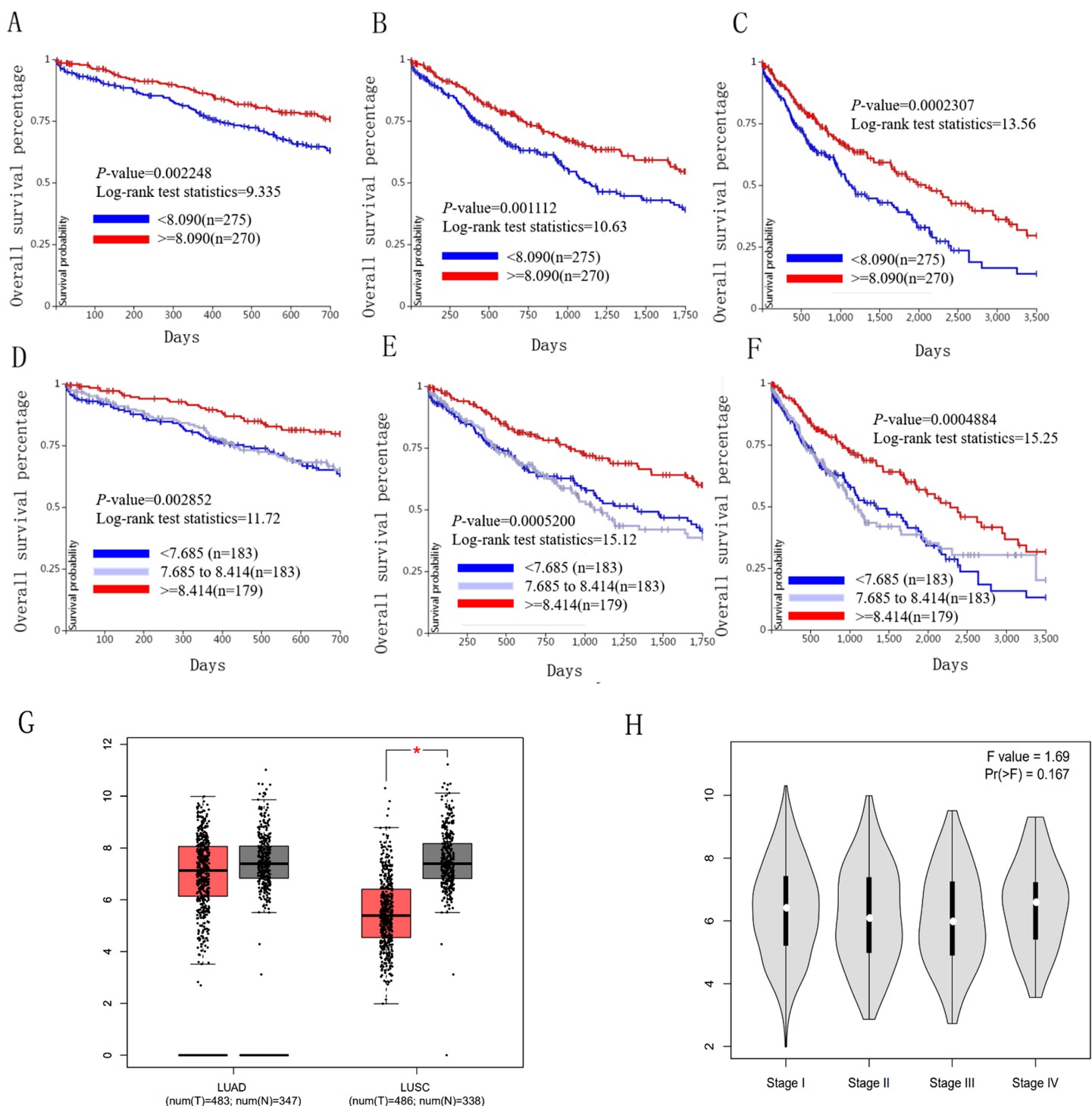

**Figure 6 Overall survival and disease-free survival analyses of different ICAM1 expression lung cancer patients.** (A–F) Overall survival and disease-free survival analyses of the ICAM1 were performed in TCGA online website. (G and H) The mRNA level of ICAM1 was evaluated in lung cancer using GEPIA analysis $p < 0.05$ was considered to indicate a statistically significant difference.

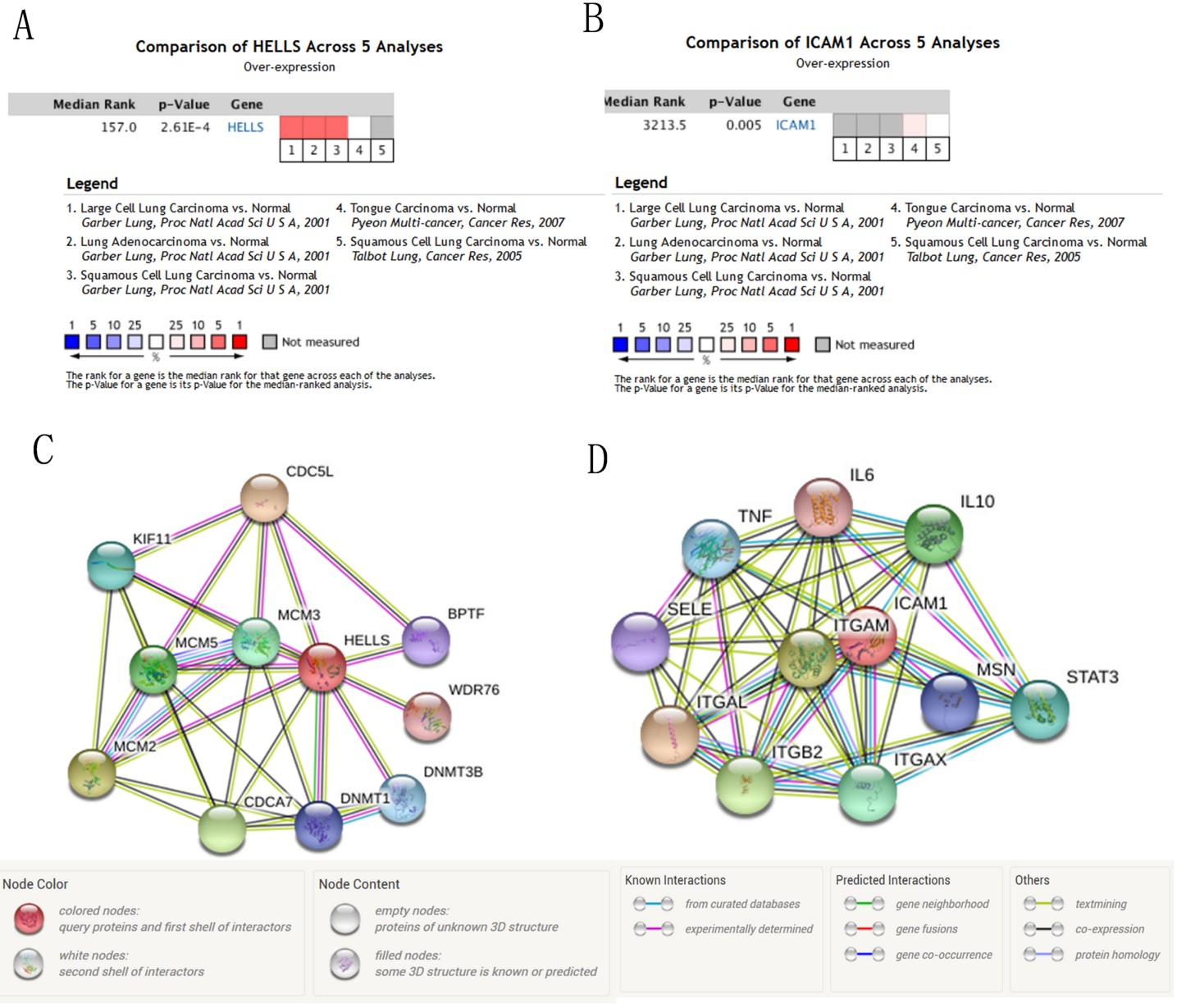

**Figure 7 The mRNA level of HELLS and ICAM1, the PPI network of HELLS and ICAM1 were constructed.** (A and B) The mRNA level of HELLS and ICAM1 were evaluated in lung cancer among four studies using ONCOMINE analysis. (C and D) PPI network of HELLS and ICAM1 were constructed by STRING database.

maintenance protein family, such as MCM5, MCM3 and MCM2 (Fig. 7C). Interactions between ICAM1 and other genes associated with inflammation were also observed in the present study (Fig. 7D). To further assess the expression of ICAM1 and HELLS, we measured mRNA levels in 79 cases of lung cancer and paired paracancer samples. RT-qPCR results showed that HELLS expression in lung cancer tissues was upregulated when compared with normal tissue, while ICAM1 expression in lung cancer tissues was downregulated when compared with normal tissue. When compared with the normal lung cancer cell line BEAS2B, HELLS mRNA levels in human lung cancer cell lines H1299,

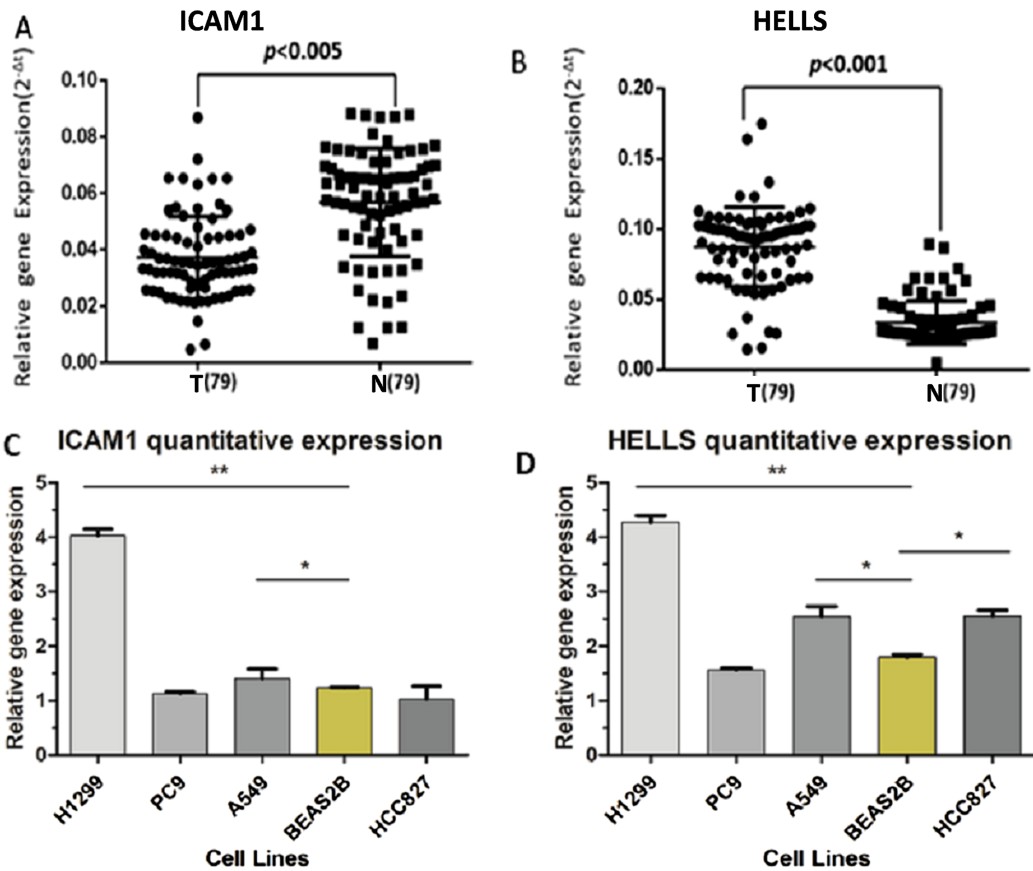

**Figure 8** **The expression levels of ICAM1 and HELLS in the lung cancer samples and the lung cancer cells.** (A and B) The expression levels of ICAM1 and HELLS in the lung cancer samples. (C and D) The expression levels of ICAM1 and HELLS in the lung cancer cells.

A549 and HCC827 were remarkably highly expressed. However, the expression of ICAM1 only decreased in HCC827 when compared with the normal lung epithelial cell line BEAS2B (Fig. 8).

# DISCUSSION

Over the past few decades, experts have explored the causes and underlying mechanisms of lung cancer formation and development through extensive basic and clinical research. However, most previous studies focused on the results of a single genetic event or single cohort study with inconsistent and incomplete results, and the incidence and mortality rate of lung cancer worldwide remain high (*Zeng et al., 2019*; *Ning et al., 2019*). Our study integrated three cohort profile datasets from different studies and used bioinformatics methods to perform an in-depth analysis, identifying 232 frequent changes in DEGS (40 up-regulations and 192 down-regulations). The 232 DEGs were then allocated to three groups according to GO terminology (molecular function, biological process and cell component) using a variety of methods. GO and KEGG analyses showed significant enrichment in these DEGs based on function and signaling pathway analysis. A PPI

network complex was developed for the DEGs to filter the hub genes and dysregulated pathways. To determine the expression pattern, potential function, and different prognostic values of the DEGS, we performed a detailed analysis of the expression of DEGS in lung cancer patients.

HELLS (also known as SMARCA6 and PASG) is the main member of the SNF2 chromatin remodeling enzyme family. The human HELLS gene is located in the c3–d1 region of chromosome 10q23–q24, while the mouse homolog is located in the same region of chromosome 19. Related studies have shown that HELLS plays an important regulatory role during normal embryonic development (*Sun et al., 2004*) and encodes a lymphoid-specific helicase. Other helicases are involved in processes involving DNA strand separation, including replication, repair, recombination, and transcription. Lymphoid-specific helicase has been shown to be involved with cellular proliferation (*Geiman, Durum & Muegge, 1998*; *Lee et al., 2000*). To maintain the DNA methylation pattern of the mammalian genome (*Myant & Stancheva, 2008*), HELLS typically interacts with DNA methyltransferases. According to recent research, HELLS also plays an important regulatory role in cell proliferation and, possibly, in the development of cancer (*He et al., 2016*; *Benavente et al., 2014*; *Teh et al., 2012*; *Tao et al., 2011*). HELLS is a key epigenetic driver of hepatocellular carcinoma (HCC) and inhibits multiple tumor suppressor genes by promoting the occupancy of nucleosomes of NFR and enhancers to promote HCC progression (*Law et al., 2019*). Recent studies have shown that HELLS genes are upregulated in nasopharyngeal carcinoma, retinoblastoma, head and neck cancer, and breast cancer. However, the detailed mechanisms of HELLS in cancer, particularly the reasons for its differential expression and downstream targets, need further research. The present study screened several DEGs in three datasets to reveal that increased levels of HELLS mRNA were significantly associated with poor OS in lung cancer patients, suggesting that HELLS may be a potential novel predictor of prognosis.

Intercellular adhesion molecule 1 (ICAM1) is an important member of the immunoglobulin superfamily. It is a glycosylated transmembrane protein that plays a key role in immune synapse formation, T cell activation, leukocyte trafficking, and various cellular immune responses. A large number of studies have shown that ICAM1 shows higher expression in mesenchymal stem cells such as bone marrow, placenta, fat and periodontal ligament (*Brooke et al., 2008*; *De Francesco et al., 2009*; *Sununliganon & Singhatanadgit, 2012*). Studies have also shown that ICAM-1 is a marker of human and mouse liver cancer stem cells and is involved in the metastasis of liver cancer cells. Its expression is regulated by the stem cell transcription factor Nanog (*Liu et al., 2013*). Reduced expression of ICAM-1 could play a role in the suppression of tumor progression in many cancer cells, such as breast cancer (*Ogawa et al., 1998*), gastric cancer (*Fujihara et al., 1999*), lung cancer (*Kotteas et al., 2014*) and colorectal cancer (*Maeda et al., 2002*). Additionally, ICAM1 and CD44 may have compensatory effects to maintain the dry characteristics of esophageal squamous cell carcinoma, indicating multiple targeted therapies that can be combined and considered in cancer treatment (*Tsai et al., 2015*). Research has demonstrated that ICAM1 is involved in angiogenesis through the regulation of endothelial cell migration (*Kevil et al., 2004*). Additional studies have shown that

ICAM-1 in the systemic circulation of lung cancer patients can bind to leukocyte-function associated antigen-1 of cytotoxic lymphocytes in the blood, enabling cancer cells to evade immune recognition mechanisms (*Kim et al., 2017*). Other studies have shown that cannabinoid-induced ICAM-1 can increase LAK cell-mediated tumor cell killing ability in lung cancer, a novel antitumor mechanism of cannabinoids (*Haustein et al., 2014*). From these three datasets, we identified that ICAM-1 is dysregulated in lung cancer. In combination with a series of previous studies, we found that decreased ICAM1 mRNA levels predict poor prognoses in patients with lung cancer (*Melis et al., 1996*; *Haustein et al., 2014*; *Schellhorn et al., 2015*). Therefore, ICAM1 may be a novel potential therapy target for lung cancer patients.

## CONCLUSIONS

In summary, we analyzed multiple cohort datasets and integrated bioinformatics to identify and screen 232 candidate genes, and we constructed a PIP network complex to screen 129 gene nodes and 10 node degree genes in DEGs. We found that elevated HELLS and decreased ICAM1 mRNA levels are predictive of poor prognoses in lung cancer patients, which could significantly improve our understanding of the causes and potential molecular events of lung cancer. However, our findings should be supplemented, and the direction for further research may include related mechanism validation studies. Whether the selected molecules have clinical significance should be verified and discussed. Therefore, further research is required to clarify the exact molecular mechanisms of these genes in lung cancer.

### Funding
The authors received no funding for this work.

### Competing Interests
The authors declare that they have no competing interests.

### Author Contributions
- Wei Zhu conceived and designed the experiments, analyzed the data, prepared figures and/or tables, authored or reviewed drafts of the paper, and approved the final draft.
- Lin Lin Li conceived and designed the experiments, analyzed the data, prepared figures and/or tables, authored or reviewed drafts of the paper, and approved the final draft.
- Yiyan Songyang performed the experiments, prepared figures and/or tables, and approved the final draft.
- Zhan Shi analyzed the data, authored or reviewed drafts of the paper, and approved the final draft.
- Dejia Li conceived and designed the experiments, prepared figures and/or tables, and approved the final draft.

## Human Ethics

The following information was supplied relating to ethical approvals (i.e., approving body and any reference numbers):

The Ethics Committee of the Medical College of Wuhan University granted ethical approval to carry out the study within its facilities (ethical Application Ref: 2018001).

## Data Availability

Data is available at NCBI GEO: GSE18842, GSE19188 and GSE27262.

The raw data is available in the Supplemental Files.

## Supplemental Information

Supplemental information for this article can be found online at http://dx.doi.org/10.7717/peerj.8731#supplemental-information.

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

# PeerJ

**Gao J, Aksoy BA, Dogrusoz U, Dresdner G, Gross B, Sumer SO, Sun Y, Jacobsen A, Sinha R, Larsson E, Cerami E, Sander C, Schultz N. 2013.** Integrative analysis of complex cancer genomics and clinical profiles using the cBioPortal. *Science Signaling* **6(269)**:pl1 DOI 10.1126/scisignal.2004088.

**Geiman TM, Durum SK, Muegge K. 1998.** Characterization of gene expression, genomic structure, and chromosomal localization of Hells (Lsh). *Genomics* **54(3)**:477–483 DOI 10.1006/geno.1998.5557.

**Haeussler M, Zweig AS, Tyner C, Speir ML, Rosenbloom KR, Raney BJ, Lee CM, Lee BT, Hinrichs AS, Gonzalez JN, Gibson D, Diekhans M, Clawson H, Casper J, Barber GP, Haussler D, Kuhn RM, Kent WJ. 2019.** The UCSC Genome Browser database: 2019 update. *Nucleic Acids Research* **47(D1)**:D853–D858 DOI 10.1093/nar/gky1095.

**Haustein M, Ramer R, Linnebacher M, Manda K, Hinz B. 2014.** Cannabinoids increase lung cancer cell lysis by lymphokine-activated killer cells via upregulation of ICAM-1. *Biochemical Pharmacology* **92(2)**:312–325 DOI 10.1016/j.bcp.2014.07.014.

**He X, Yan B, Liu S, Jia J, Lai W, Xin X, Tang C-E, Luo D, Tan T, Jiang Y, Shi Y, Liu Y, Xiao D, Chen L, Liu S, Mao C, Yin G, Cheng Y, Fan J, Cao Y, Muegge K, Tao Y. 2016.** Chromatin remodeling factor LSH drives cancer progression by suppressing the activity of fumarate hydratase. *Cancer Research* **76(19)**:5743–5755 DOI 10.1158/0008-5472.CAN-16-0268.

**Hou J, Aerts J, Den Hamer B, Van Ijcken W, Den Bakker M, Riegman P, Van der Leest C, Van der Spek P, Foekens JA, Hoogsteden HC, Grosveld F, Philipsen S. 2010.** Gene expression-based classification of non-small cell lung carcinomas and survival prediction. *PLOS ONE* **5(4)**:e10312 DOI 10.1371/journal.pone.0010312.

**Kevil CG, Orr AW, Langston W, Mickett K, Murphy-Ullrich J, Patel RP, Kucik DF, Bullard DC. 2004.** Intercellular adhesion molecule-1 (ICAM-1) regulates endothelial cell motility through a nitric oxide-dependent pathway. *Journal of Biological Chemistry* **279(18)**:19230–19238 DOI 10.1074/jbc.M312025200.

**Kim E, Kim W, Lee S, Chun J, Kang J, Park G, Han I, Yang HJ, Youn H, Youn B. 2017.** TRAF4 promotes lung cancer aggressiveness by modulating tumor microenvironment in normal fibroblasts. *Scientific Reports* **7(1)**:8923 DOI 10.1038/s41598-017-09447-z.

**Kohl M, Wiese S, Warscheid B. 2011.** Cytoscape: software for visualization and analysis of biological networks. *Methods in Molecular Biology* **696**:291–303 DOI 10.1007/978-1-60761-987-1_18.

**Kotteas EA, Boulas P, Gkiozos I, Tsagkouli S, Tsoukalas G, Syrigos KN. 2014.** The intercellular cell adhesion molecule-1 (ICAM-1) in lung cancer: implications for disease progression and prognosis. *Anticancer Research* **34**:4665–4672.

**Latendresse M, Paley S, Karp PD. 2012.** Browsing metabolic and regulatory networks with BioCyc. *Methods in Molecular Biology* **804(11)**:197–216 DOI 10.1007/978-1-61779-361-5_11.

**Law C-T, Wei L, Tsang FH-C, Chan CY-K, Xu IM-J, Lai RK-H, Ho DW-H, Lee JM-F, Wong CC-L, Ng IO-L, Wong C-M. 2019.** HELLS regulates chromatin remodeling and epigenetic silencing of multiple tumor suppressor genes in human hepatocellular carcinoma. *Hepatology* **69(5)**:2013–2030 DOI 10.1002/hep.30414.

**Lebrec JJP, Huizinga TWJ, Toes REM, Houwing-Duistermaat JJ, Van Houwelingen HC. 2009.** Integration of gene ontology pathways with North American Rheumatoid Arthritis Consortium genome-wide association data via linear modeling. *BMC Proceedings* **3(S7)**:S94 DOI 10.1186/1753-6561-3-S7-S94.

Lee DW, Zhang K, Ning ZQ, Raabe EH, Tintner S, Wieland R, Wilkins BJ, Kim JM, Blough RI, Arceci RJ. 2000. Proliferation-associated SNF2-like gene (PASG): a SNF2 family member altered in leukemia. *Cancer Research* **60**:3612–3622.

Liu S, Li N, Yu X, Xiao X, Cheng K, Hu J, Wang J, Zhang D, Cheng S, Liu S. 2013. Expression of intercellular adhesion molecule 1 by hepatocellular carcinoma stem cells and circulating tumor cells. *Gastroenterology* **144(5)**:1031–1041 DOI 10.1053/j.gastro.2013.01.046.

Maeda K, Kang SM, Sawada T, Nishiguchi Y, Yashiro M, Ogawa Y, Ohira M, Ishikawa T, Hirakawa-YS CK. 2002. Expression of intercellular adhesion molecule-1 and prognosis in colorectal cancer. *Oncology Reports* **9**:511–514.

Melis M, Spatafora M, Melodia A, Pace E, Gjomarkaj M, Merendino AM, Bonsignore G. 1996. ICAM-1 expression by lung cancer cell lines: effects of upregulation by cytokines on the interaction with LAK cells. *European Respiratory Journal* **9(9)**:1831–1838 DOI 10.1183/09031936.96.09091831.

Myant K, Stancheva I. 2008. LSH cooperates with DNA methyltransferases to repress transcription. *Molecular and Cellular Biology* **28(1)**:215–226 DOI 10.1128/MCB.01073-07.

Ning Y, Liu W, Guan X, Xie X, Zhang Y. 2019. CPSF3 is a promising prognostic biomarker and predicts recurrence of non-small cell lung cancer. *Oncology Letters* **18**:2835–2844 DOI 10.3892/ol.2019.10659.

Ogawa Y, Hirakawa K, Nakata B, Fujihara T, Sawada T, Kato Y, Yoshikawa K, Sowa M. 1998. Expression of intercellular adhesion molecule-1 in invasive breast cancer reflects low growth potential, negative lymph node involvement, and good prognosis. *Clinical Cancer Research* **4**:31–36.

Rhodes DR, Yu J, Shanker K, Deshpande N, Varambally R, Ghosh D, Barrette T, Pandey A, Chinnaiyan AM. 2004. ONCOMINE: a cancer microarray database and integrated data-mining platform. *Neoplasia* **6(1)**:1–6 DOI 10.1016/S1476-5586(04)80047-2.

Sanchez-Palencia A, Gomez-Morales M, Gomez-Capilla JA, Pedraza V, Boyero L, Rosell R, Farez-Vidal ME. 2011. Gene expression profiling reveals novel biomarkers in nonsmall cell lung cancer. *International Journal of Cancer* **129(2)**:355–364 DOI 10.1002/ijc.25704.

Schellhorn M, Haustein M, Frank M, Linnebacher M, Hinz B. 2015. Celecoxib increases lung cancer cell lysis by lymphokine-activated killer cells via upregulation of ICAM-1. *Oncotarget* **6(36)**:39342–39356 DOI 10.18632/oncotarget.5745.

Siegel RL, Miller KD, Jemal A. 2016. Cancer statistics, 2016. *CA: A Cancer Journal for Clinicians* **66(1)**:7–30 DOI 10.3322/caac.21332.

Siegel RL, Miller KD, Jemal A. 2018. Cancer statistics, 2018. *CA: A Cancer Journal for Clinicians* **68(1)**:7–30 DOI 10.3322/caac.21442.

Sun LQ, Lee DW, Zhang Q, Xiao W, Raabe EH, Meeker A, Miao D, Huso DL, Arceci RJ. 2004. Growth retardation and premature aging phenotypes in mice with disruption of the SNF2-like gene, PASG. *Genes & Development* **18(9)**:1035–1046 DOI 10.1101/gad.1176104.

Sununliganon L, Singhatanadgit W. 2012. Highly osteogenic PDL stem cell clones specifically express elevated levels of ICAM1, ITGB1 and TERT. *Cytotechnology* **64(1)**:53–63 DOI 10.1007/s10616-011-9390-5.

Szklarczyk D, Gable AL, Lyon D, Junge A, Wyder S, Huerta-Cepas J, Simonovic M, Doncheva NT, Morris JH, Bork P, Jensen LJ, Von Mering C. 2019. STRING v11: protein–protein association networks with increased coverage, supporting functional discovery in genome-wide experimental datasets. *Nucleic Acids Research* **47(D1)**:D607–D613 DOI 10.1093/nar/gky1131.

**Tang Z, Li C, Kang B, Gao G, Li C, Zhang Z. 2017.** GEPIA: a web server for cancer and normal gene expression profiling and interactive analyses. *Nucleic Acids Research* **45(W1)**:W98–W102 DOI 10.1093/nar/gkx247.

**Tao Y, Liu S, Briones V, Geiman TM, Muegge K. 2011.** Treatment of breast cancer cells with DNA demethylating agents leads to a release of Pol II stalling at genes with DNA-hypermethylated regions upstream of TSS. *Nucleic Acids Research* **39(22)**:9508–9520 DOI 10.1093/nar/gkr611.

**Teh M-T, Gemenetzidis E, Patel D, Tariq R, Nadir A, Bahta AW, Waseem A, Hutchison IL. 2012.** FOXM1 induces a global methylation signature that mimics the cancer epigenome in head and neck squamous cell carcinoma. *PLOS ONE* **7(3)**:e34329 DOI 10.1371/journal.pone.0034329.

**The Cancer Genome Atlas Network. 2012.** Comprehensive molecular portraits of human breast tumours. *Nature* **490(7418)**:61–70 DOI 10.1038/nature11412.

**The Gene Ontology Consortium. 2015.** Gene Ontology Consortium: going forward. *Nucleic Acids Research* **43(D1)**:D1049–D1056 DOI 10.1093/nar/gku1179.

**Tsai ST, Wang PJ, Liou NJ, Lin PS, Chen CH, Chang WC. 2015.** ICAM1 is a potential cancer stem cell marker of esophageal squamous cell carcinoma. *PLOS ONE* **10(11)**:e0142834 DOI 10.1371/journal.pone.0142834.

**Vogelstein B, Papadopoulos N, Velculescu VE, Zhou S, Diaz LA, Kinzler KW. 2013.** Cancer genome landscapes. *SCIENCE* **339(6127)**:1546–1558 DOI 10.1126/science.1235122.

**Wei T-YW, Hsia J-Y, Chiu S-C, Su L-J, Juan C-C, Lee Y-CG, Chen J-MM, Chou H-Y, Huang J-Y, Huang H-M, Yu C-TR. 2014.** Methylosome protein 50 promotes androgen- and estrogen-independent tumorigenesis. *Cellular Signalling* **26(12)**:2940–2950 DOI 10.1016/j.cellsig.2014.09.014.

**Wei T-YW, Juan C-C, Hisa J-Y, Su L-J, Lee Y-CG, Chou H-Y, Chen J-MM, Wu Y-C, Chiu S-C, Hsu C-P, Liu K-L, Yu C-TR. 2012.** Protein arginine methyltransferase 5 is a potential oncoprotein that upregulates G1 cyclins/cyclin-dependent kinases and the phosphoinositide 3-kinase/AKT signaling cascade. *Cancer Science* **103(9)**:1640–1650 DOI 10.1111/j.1349-7006.2012.02367.x.

**Zeng T, Chen C, Yang P, Zuo W, Liu X, Zhang Y. 2019.** A protective role for RHOJ in nonsmall cell lung cancer based on integrated bioinformatics analysis. Epub ahead of print 23 October 2019. *Journal of Computational Biology* DOI 10.1089/cmb.2019.0209.

**Zhou Y, Zhou B, Pache L, Chang M, Khodabakhshi AH, Tanaseichuk O, Benner C, Chanda SK. 2019.** Metascape provides a biologist-oriented resource for the analysis of systems-level datasets. *Nature Communications* **10(1)**:1523 DOI 10.1038/s41467-019-09234-6.