# Peer review of "Identification and validation of HELLS (Helicase, Lymphoid-Specific) and ICAM1 (Intercellular adhesion molecule 1) as potential diagnostic biomarkers of lung cancer"

_PeerJ, doi:10.7717/peerj.8731_

## Round 0.1 · original submission · Minor Revisions

All issues indicated by the reviewers should be fixed and the manuscript should be revised accordingly

Reviewer 1 ·

Basic reporting

Wei Zhu et al. used integrated bioinformatics analysis to identify two genes, HELLS and ICAM1, involved in lung cancer. Lung cancer patients showed increased and decreased mRNA expression for HELLS and ICAM1, respectively. These two genes could be used to predict lung cancer as well as can serve as potential therapeutic targets.

The structure of the article conforms to the PeerJ standards. Clear and professional English was used in the manuscript. The introduction provides an appropriate background. The literature is relevant. The quality of the figures requires improvement. The raw data were supplied.

Experimental design

The manuscript clearly stated how the research fills critical knowledge gaps. The research questions are well defined. The methods were described with sufficient details.

Validity of the findings

Data was interpreted completely and supported the conclusion. The authors successfully established that HELLS and ICAM1 could be two essential genes related to lung cancer.

Additional comments

- Please provide an English translation version of the informed consent and any other documents in the supplementary data.
- Texts inside the figure 1B, 3B, 4B, 7A, 7B, 7C, 7D are hard to read. Please use the bigger font size.

Reviewer 2 ·

Basic reporting

Zhu et. al. used bioinformatics, systems biology approaches coupled with experimental analyses involving lung cancer samples, cell culture and RT-PCR to initially identify differentially expressed genes and candidate genes in pathways related to lung cancer. Further, Zhu et. al. expand their experimental and computational analysis to show that HELLS and ICAM1 might be the key genes involved with lung cancer progression, and elevated levels of HELLS mRNA and decreased levels of ICAM1 mRNA are predictive of poor prognosis in lung cancer patients.
This study is a nice study integrating experimental and computational methods to identify and validate diagnostic biomarkers for lung cancer. The methodology used in this study can be extrapolated to other types of cancer, to identify and validate potential biomarkers robustly. Therefore, I think this is a nice and important piece of work which will attract readers working in cancer research, and which will be used by other labs working in the field of cancer research. There are some issues which will have to be addressed by the authors, as I have mentioned in my review, point-by-point. Also, the manuscript has many grammatical errors that need to be taken care of. I have done my best to point out some of them. But, I believe, there might be many more grammatical errors that need to be taken care of. If the authors can satisfy these concerns and the review points, I would be happy to see that this manuscript gets published in PeerJ.

Experimental design

No comment

Validity of the findings

No comments

Additional comments

REVIEW POINTS

Line 50, 51: “However, due to the heterogeneity of the tissue or sample in the existing studies, the results are limited or inconsistent”.
Could the authors elaborate on this statement? Also, it would help the readers if the authors provided one or two examples commenting on the shortcomings of previously published microarray data analysis and how their own method addresses these shortcomings.

Line 55:” We downloaded three original microarray datasets:”
Could the authors provide a rationale for selecting these three datasets over the other available datasets?

Line 140: Were the primers purchased or synthesized in the lab? If they were purchased, please include the name of the vendor. Primer sequences of GAPDH seems like a standard sequence that is being used. For HELLS and ICAM1, are these sequences standard to the available ones in existing literature?
Line 164: The text in the figure 1A and 1B is not readable. Could the authors include high-resolution figure that makes is easier for the reader to understand the text?

Line 169: In Table2, GO id (GO:0008285 ), the description tab does not contain the full information for that particular GO. It seems like an editing issue. Please rectify that.
Line 176: Go should be changed to GO
Line 178: The resolution of the Figure 1B should be changed, it is hard to read the legend. Since, the authors base this result on the network, a high-res figure of the network and the legend would be nice to have.
Line 183: “Using the string online database and Cytoscape software 13, a total of 232 DEGs of 232 common changes in DEGs were filtered into the PPI network complex”
This statement is confusing. Could the authors rephrase this statement?
Line 186: Could the authors explain what is meant by a “cell angle”
Line 186: “Using the Metascope to analyze the functions of the genes involved in this module is mainly focused on cell division, pre-mitotic stage and mitosis cell cycle transition”
Could the authors please rephrase this statement.
Line 192: Was the analysis of biological processes of hub genes carried out by the cBioPortal online platform in Fig. 4C?
Line 221: However, levels of ICAM1 mRNA in human lung cancer cell lines, only HCC827, was a low expression, compared with normal lung epithelial cell line, BEAS2B(Fig.8).
Rephrase this statement, it seems confusing to understand
Line 228: “However, most previous studies have focused on the results of a single genetic event or a single cohort study, and the results from these studies have been inconsistent and incomplete, the incidence and mortality of lung cancer worldwide remain high”
Please provide few references
Line 246: Please rephrase “This protein …..”
Line 257: Change “and we also revealed” to “and also revealed”
Line 260: Introduce the expanded name of the abbreviation, ICAM1, where it first appears in the manuscript.
Line 281: “Coupled with a series of previous studies, we revealed that decreased ICAM1 mRNA levels predict poor prognosis in patients with lung cancer. Therefore, ICAM1 may be a novel potential therapy target for patients with lung cancer”.
If decreased ICAM1 mRNA levels predict poor prognosis in patients with lung cancer, how can it be a novel target?
Line 292: Rephrase the statement.

Figures: Figure 1A, 1B, 3B, 5G, 6G 8A, 8B: Please provide high-res figures
Also for figures 5G, 6G, please include what the Asterix means and provide relevant statistical information

---

## Round 0.2 · accepted · Accept

All issues pointed by the reviewers were adequately addressed and the manuscript was revised accordingly. Therefore it is acceptable now.